# *Anas barbariae* 200K Modulates Cell Stiffness and Oxidative Stress in Microglial Cells In Vitro

**DOI:** 10.3390/ijms26041451

**Published:** 2025-02-09

**Authors:** Anne Paumier, Justine Verre, Gaël Runel, Julien Chlasta, Sandra Tribolo, Stéphanie Chanut

**Affiliations:** 1Laboratoires BOIRON, Research Department, 2 Avenue de l’Ouest Lyonnais, 69510 Messimy, France; anne.paumier@boiron.fr (A.P.); justine.verre@boiron.fr (J.V.); stephanie.chanut@boiron.fr (S.C.); 2BioMeca, 60C Avenue Rockfeller, 69008 Lyon, France; julien.chlasta@bio-meca.com

**Keywords:** *Anas barbariae*, cell stiffness, antioxidant, atomic force microscopy

## Abstract

*Anas barbariae* 200K, a homeopathic medicine, is traditionally used for influenza-like illnesses. We investigated the effects of *Anas barbariae* 200K on microglial cells, a subpopulation of macrophages specific to the central nervous system often used to study the inflammatory processes and oxidative stress generated during influenza-like episodes. The study demonstrates the effect of *Anas barbariae* 200K on cell stiffness and the reactive oxygen species production using atomic force microscopy and fluorescence microscopy techniques, respectively. Our results showed that *Anas barbariae* 200K rapidly increased cell stiffness in resting cells by 41% compared with the vehicle. In inflamed cells, cell stiffness was decreased by 21% when treated with *Anas barbariae* 200K compared with the vehicle. Finally, *Anas barbariae* 200K caused a reorganisation of filamentous actin, with marked relocation of actin at the cell extremities. Moreover, *Anas barbariae* 200K significantly decreased the reactive oxygen species (ROS) production in inflamed microglial cells by 40% (total intracellular ROS) and by 67% (mitochondrial ROS) compared with the vehicle. These results strongly suggest an effect of *Anas barbariae* 200K at a cellular level on cell stiffness and actin cytoskeleton. This sheds light on the biological mechanism of action of this homeopathic preparation.

## 1. Introduction

Homeopathy is a widely used complementary medicine that has been practised for over 200 years. Homeopathic medicines have been used in the prevention and treatment of infectious diseases [1,2,3,4]. Oscillococcinum^®^ is traditionally used in the relief of influenza-like symptoms such as fever, headache, chills, and body aches. Patients with flu symptoms have shown a decline in symptoms 48 h after taking this homeopathic medicine [5,6]. Clinical studies show its efficacy in a population of Chronic Obstructive Pulmonary Disease (COPD) patients by reducing the frequency and the duration of Upper Respiratory Tract Infections (URTIs) during an outbreak of influenza [7,8]. In some cases, the severe form of infection by the influenza virus is accompanied by a cytokine/chemokine storm caused by an inflammatory response in the lungs [9]. Cytokines are proteins secreted by cells that are essential for the communication between cells in the immune system and the resolution of infectious diseases. When the cytokines are intensively produced and uncontrolled, a cytokine storm is created, which causes damage to the cells and organs. This inflammatory state is closely related to oxidative stress. Under physiological conditions, the reactive oxygen species (ROS) are essential to regulate cell proliferation, migration, and apoptosis with the activation of various signalling pathways. An imbalance between the production of ROS and the antioxidant molecules leads to an increased level of oxidative stress. The over-production of ROS enhances the inflammatory state and damages biomolecules (proteins, nucleic acids, lipids), cell membranes, and organelles especially mitochondria [10,11].

Cellular mechanical properties, in particular stiffness, are critical for cellular processes such as migration, protrusion, division, and morphology [12]. Actin filaments, part of the cytoskeleton, serve as the backbone for the intracellular force transmission and transduction of external mechanical signals via focal adhesion complexes [13]. The assessment of cellular mechanical properties at a subcellular resolution is crucial for understanding these processes, with atomic force microscopy (AFM) being a commonly used approach [14].

Microglial cells are the macrophage population present in the central nervous system. In a healthy brain, microglia are found in their ramified morphology, acting as sentinels, and maintaining homeostasis. They are activated by various stimuli caused by brain injury or disease [15,16]. When the cells are in an activated state, their morphology changes, and they move towards the damaged site via chemotaxis. It is clear that microglial cells play a key role in the pathophysiology of neurodegeneration [17]. A brain injury results in a cascade of cellular responses which produce neuroinflammation. In rats, distinct subpopulations of microglial cells move directly to the damaged site [18]. Primary microglial cells are good candidates for studying oxidative stress and homeopathic drugs because they are susceptible and reactive cells [19].

In the present study, we evaluated the activity of *Anas barbariae* 200K dilution in vitro in murine microglial cells focusing on the cell surface ultrastructure, the reorganisation of the cytoskeleton, and the antioxidant properties using atomic force microscopy and fluorescence microscopy techniques. This comprehensive approach promises to provide insights into the therapeutic potential and mechanistic actions of *Anas barbariae* 200K in modulating inflammatory responses.

## 2. Results

### 2.1. Treatments Induce Time-Dependent Changes in Morphology

In resting microglial cells (CTRLs), we first evaluated the reorganisation of the actin cytoskeleton with different treatments over time. The vehicle (sterile water) treatment showed no significant variation throughout the experiment, suggesting that the addition of solutions per se did not notably impact the cell morphology (Figure 1A,B). Jasplakinolide (JPK), a cyclo-depsipeptide, a well-known actin filament polymerising and stabilising agent was used as a positive control of increased stiffness. Following the addition of JPK, a significant increase in the aspect ratio was observed compared with the vehicle group at both T2min and T4min time points by 25% and 151%, respectively. Treatment with *Anas barbariae* (*A. barbariae*) 200K exhibited a significant increase of 41% at T4min in the aspect ratio compared with the vehicle, but no significant changes were observed at T2min. Notably, no significant changes in the aspect ratio were observed at T20min for any of the treatments.

### 2.2. Increase in Stiffness Following Short-Term Treatments

Secondly, we investigated the impact of *A. barbariae* 200K on the cell surface ultrastructure of resting microglial cells using atomic force microscopy (AFM). This advanced technique enables the measurement of mechanical properties, such as the elastic modulus (Ea) of biological materials, including cells, at high resolution and under quasi-physiological conditions in their culture medium. By monitoring the changes in properties before and after the addition of drugs, AFM offers valuable insights into cellular responses. In our study, we compared the effects of *A. barbariae* 200K under different conditions: resting control cells (CTRLs) with a cell culture medium and cells treated with the vehicle or JPK. Our observations revealed distinct behaviours depending on the conditions tested. All the comparisons showed significant differences compared with the CTRLs, except for the vehicle condition after 1 min. Upon the addition of *A. barbariae* 200K and JPK, we observed a rapid increase in elastic modulus indicating enhanced cell stiffness. Notably, *A. barbariae* 200K induced a significant increase of 51% compared with the vehicle alone (Figure 1C). To further analyse the cellular response, we evaluated the proportion of cells reacting to the treatments. Interestingly, over 80% of the cells treated with JPK and *A. barbariae* 200K exhibited an increase in elastic Young’s modulus, compared to approximately 40% in the CTRL and vehicle groups (Figure 1D). These results suggest that cells promptly respond to *A. barbariae* 200K treatment by increasing their stiffness, highlighting the potential mechanistic involvement of *A. barbariae* 200K in cellular processes affecting the mechanical properties.

### 2.3. Temporal Dynamics of the Cellular Response

Our findings have convincingly demonstrated that cells exhibit an immediate physical response upon the addition of *A. barbariae* 200K. To further explore this phenomenon, we investigated the longer-term effects of *A. barbariae* 200K on the cellular behaviour. We monitored the mechanical properties of cells over a period of 20 min (Figure 1E), after the addition of JPK, *A. barbariae* 200K, the vehicle, or the medium alone. At 1 min post-addition, we observed a rapid increase in cell stiffness in response to *A. barbariae* 200K, in stark contrast to the CTRL and vehicle conditions. Interestingly, however, after this initial rise, the cells treated with *A. barbariae* 200K began to soften after 2 min, returning almost to their initial mechanical state. The vehicle-treated cells and CTRL-treated cells exhibited similar temporal patterns, with a notable increase in the cell stiffness observed only in the vehicle-treated cells between 2 and 4 min. Interestingly, the treatment with JPK induced fluctuations in cell stiffness throughout the duration of the experiment. These results suggest a dynamic response of resting microglial cells to *A. barbariae* 200K, characterised by a rapid increase in stiffness followed by subsequent softening, indicating a complex interplay between the treatment and cellular mechanics over time.

### 2.4. A. barbariae 200K Induces Stiffness Reduction in Inflamed Cells

We then investigated the behaviour of inflamed cells. Firstly, we aimed to observe the mechanical differences in the cells treated with lipopolysaccharide (LPS), a bacterial endotoxin, for various durations. Here, we sought to characterise the general increase in the cell stiffness of about 32% compared with the baseline (T0). We observed an increase of 32% from 2 h. At 4 h and 6 h, the stiffness was slightly lower compared with 2 h, about 17%, reaching a maximum at 8 h and was maintained at 48% until 24 h (Figure 2A). Subsequently, we chose a stimulation period of 4 h to observe an increase in the stiffness with LPS and a decrease in stiffness following *A. barbariae* 200K treatment. To determine the most appropriate treatment duration to observe changes in the cellular mechanical parameters, we administered *A. barbariae* 200K after 4 h of incubation with LPS and measured the cell stiffness at different time points. Given the treatment’s short and cyclical action, we chose a treatment duration of 4 min resulting in a significant stiffness decrease (30%) (Figure 2B). After the validation of the LPS incubation time and treatment duration, we examined the effect of treatments on the mechanical behaviour of the inflamed cells. Dexamethasone (DEX), a corticosteroid medication used to treat a variety of conditions, including inflammatory disorders, was used as a positive control. The addition of DEX resulted in a significant reduction in the cell stiffness of 37% compared with LPS. These data validated the relevance of the mechanical analysis of the cells under inflammatory conditions. Additionally, the addition of the vehicle did not show any significant difference. Finally, *A. barbariae* 200K treatment showed a decrease in stiffness of 21% compared with the vehicle, similar to that observed with DEX showing a 28% decrease compared with the vehicle (Figure 2C). To further analyse the cellular response, we evaluated the proportion of inflamed cells reacting to the treatments. Interestingly, about 83% of the cells treated with DEX and *A. barbariae* 200K exhibited a decrease in the elastic modulus of approximately 58% compared with the vehicle (Figure 2D). These results suggest that LPS-inflamed cells respond promptly to *A. barbariae* 200K treatment by decreasing their stiffness.

### 2.5. A. barbariae 200K Enables Actin Filament Re-Localisation

Finally, given that mechanical changes correlated with actin cytoskeleton remodelling, we aimed to ascertain the impact of the treatment’s influence on this process. As previously observed, we noted changes in the aspect ratio of non-activated cells. Here, we sought to highlight the cytoskeletal modifications in an inflamed context post-treatment. To achieve this, we examined the localisation of filamentous actin within microglial cells under different conditions. We performed the intensity profiling of the actin filaments across the cell, including through the nucleus. By comparing the actin filament localisation at T0min and T4min, we were able to assess the changes over time (Figure 3A). Specifically, the CTRL, LPS, DEX, and vehicle treatments exhibited a similar profile at both time points, with a higher intensity at the cell centre. Interestingly, only *A. barbariae* 200K treatment showed a clear profile difference with a more pronounced redistribution of filamentous actin towards its extremities (Figure 3B,B’).

### 2.6. A. barbariae 200K Reduces Oxidative Stress

Firstly, the viability of the LPS-inflamed cells was checked using various amounts of *A. barbariae* 200K at 1%, 3%, 6%, and 12% (*v*/*v*) or vehicle sterile water at the same percentages. All the incubations did not affect the viability of the cells as shown in Appendix A.

Secondly, the ROS production (measured using a CellROX DeepRed probe) was determined over a range of amount of *A. barbariae* 200K or the vehicle. This dose-dependent response allowed us to select *A. barbariae* 200K 6% (*v*/*v*) to obtain the optimal effect on the ROS production compared with the vehicle (see Appendix A).

To further study the link between ROS and the actin cytoskeleton with *A. barbariae* 200K on the LPS-inflamed cells, the oxidative stress level was evaluated by measuring the ROS production using two different probes: CellROX DeepRed for the total intracellular ROS and MitoSOX Red for the mitochondrial ROS (Figure 4). The use of the two probes allowed us to follow the pathway of the production of ROS in the microglial cells. The cells were incubated with 10 ng/mL LPS for 4 h and then co-incubated with different treatments for an additional 2 h or 16 h. The total intracellular ROS produced in the cells was measured after 6 h (Figure 4A) and 20 h (Figure 4B) of the LPS treatment. At both time points, LPS increased the total ROS production compared with a low amount of ROS detected in untreated cells. When the inflamed cells were treated with 0.63 µM DEX, the ROS production was significantly inhibited by 55% (Figure 4A) and 68% (Figure 4B) compared with LPS. DEX was used as a positive control for decreasing the ROS production in our cell model. However, there was no effect of the vehicle (sterile water) compared to LPS. Interestingly, *A. barbariae* 200K significantly decreased the total intracellular ROS by around 40% compared with the vehicle.

To explore further the antioxidant effect of *A. barbariae* 200K and to investigate the pathway of the ROS production in microglial cells, we also used the MitoSOX probe to measure the ROS production in mitochondria only (Figure 4C). The inflammation was induced with 10 ng/mL LPS for 4 h and then 0.63 µM DEX, and the vehicle or *A. barbariae* 200K was co-incubated for an additional 2 h. The mitochondrial ROS produced in the cells was measured after 6 h (Figure 4C) of LPS treatment. LPS enhanced the mitochondrial ROS production compared with the untreated cells. DEX showed a significant decrease of 62% of the ROS compared with LPS. As obtained with the CellROX, no effect of the vehicle compared to the inflamed condition was observed. *A. barbariae* 200K significantly inhibited the ROS production in the mitochondria by 67% compared with the vehicle. All these results showed that after 6 h of LPS treatment, the total intracellular ROS were mostly found in the mitochondria.

*A. barbariae* 200K rapidly reduced the total intracellular ROS and mitochondrial ROS in the LPS-inflamed cells. Its effect on the total intracellular ROS was maintained for a longer period. *A. barbariae* 200K acted as an antioxidant.

## 3. Discussion

The results of our study shed light on the potential therapeutic effects of *A. barbariae* 200K in modulating the inflammatory responses, particularly in the context of microglial cell mechanics. In resting microglial cells, our results suggest that *A. barbariae* 200K induces a rapid increase in the cell stiffness, followed by a subsequent softening phase. This dynamic response highlights the complex interplay between *A. barbariae* 200K treatment and the cellular mechanics over time. In the present study for ROS assessment, the microglial cells were firstly inflamed with LPS alone for 4 h and then treated with *A. barbariae* 200K (CellROX experiments). In our previous study [20], the microglial cells were treated with *A. barbariae* 200K before the inflammation with LPS. The order of the treatments was irrelevant, as the effect of *A. barbariae* 200K still remained the same. Regarding the stiffness, the results showed that *A. barbariae* 200K increased the cell stiffness of resting cells and decreased it in inflamed cells. In Runel et al. [20], we only investigated resting cells, and the same effect on the cell stiffness was found. Although different production batches of *A. barbariae* 200K were used in both studies, we observed similar effects on the ROS production and cell stiffness. This present study reinforces and complements our previous results reported in Runel et al. [20].

The increase in cell stiffness following *A. barbariae* 200K treatment aligns with its known immunomodulatory properties. Previous studies have implicated microglial activation and subsequent inflammatory responses in the pathogenesis of several neurological diseases [21]. LPS-induced immune activation led to changes in traction forces and increased the migration [22]. Our results suggest that *A. barbariae* 200K may exert its therapeutic effects by modulating the mechanical properties of microglial cells, potentially influencing their activation state and inflammatory phenotype. AFM is a recognised method to study the anti-inflammatory effects of drugs on inflamed macrophages. The authors demonstrated the morphological and ultrastructural change of the cells with LPS, DEX or quercetin using topography images [23].

Interestingly, our results also demonstrate that *A. barbariae* 200K treatment leads to cytoskeletal remodelling, as evidenced by the redistribution of filamentous actin towards the cellular extremities. This finding supports the notion that *A. barbariae* 200K affects the morphology and function of microglial cells, potentially influencing their ability to respond to inflammatory stimuli. Fuselier et al. showed a reduction in stiffness in response to Ruta graveolens 9CH dilution using B16F10 melanoma cells [24]. The stiffness behaviour of the cells with the treatment is highly specific regarding the cell type and the homeopathic preparation.

We also showed that *A. barbariae* 200K decreased the total intracellular ROS production in LPS-inflamed microglial cells for both short and longer periods. This antioxidant effect was previously described with other homeopathic preparations [25,26]. Moreover, the antioxidant properties of drugs have been well documented in this primary cell model [27].

Some studies have shown that LPS-inflamed microglial cells reduced significantly the ROS production in the presence of cannabinoids [28,29]. Knowing that ROS are mainly generated by mitochondria in cells [30], we also found a rapid ROS production located in the mitochondria in microglial cells. It is well described that ROS and an actin cytoskeleton are closely linked. The production of ROS affects actin polymerisation and acts as a cellular defence mechanism [31]. One hypothesis is that the decrease in ROS in inflamed microglial cells with *A. barbariae* 200K could be explained by this actin reorganisation. Another possibility is the involvement of the extracellular vesicles (EVs) secreted by the cells with antioxidant capacity. These EVs contain mRNA, miRNA, and proteins and they can carry different antioxidant enzymes such as superoxide dismutase (SOD) and catalase, as well as reactive oxygen molecules, thereby regulating oxidative stress [32,33,34]. It would be interesting to analyse the contents of EVs secreted by the inflamed microglial cells in response to homeopathic products, as the mechanism of action of homeopathic dilutions is currently not well understood.

Moreover, our study provides insights into the temporal dynamics of *A. barbariae* 200K treatment on the cellular mechanics. The observed initial increase in the cell stiffness followed by softening suggests a biphasic response to *A. barbariae* 200K, which may have important implications for its therapeutic efficacy. Moreover, *A. barbariae* 200K treatment redistributed filamentous actin towards inflamed cells’ extremities. In inflamed microglial cells, the redistribution of actin is well described, and this shift of actin is correlated to a change in microglial behaviour. The cytoskeleton rearrangement via actin polymerisation and ROS have been shown to influence phagocytosis [35,36,37]. Further investigation into the underlying molecular mechanisms driving these dynamic changes is warranted to fully elucidate the therapeutic potential of *A. barbariae* 200K in diseases.

In conclusion, our study highlights the potential of *A. barbariae* 200K as a therapeutic agent for modulating inflammatory responses. The mechanism of action is not yet well identified for homeopathic preparations. However, in the literature, there are several pieces of scientific evidence that homeopathic preparations have an effect on cell function such as oxidative stress, cell stiffness modulation and membrane organisation, neurite outgrowth, bioenergetic, gene expression, DNA damage, and inflammation [24,25,26,38,39,40,41]. Our future studies will explore the molecular pathways involved in *A. barbariae* 200K-mediated immunomodulation.

## 4. Materials and Methods

### 4.1. Homeopathic Solutions

The homeopathic solutions were prepared by Laboratoires Boiron (Messimy, France) in compliance with the European Pharmacopoeia (Ph. Eur.) monographs according to the monograph 2371 and 1038 [42,43]. *Anas barbariae* 200K (*A. barbariae* 200K) is a patented preparation produced by Laboratoires Boiron. It is Hepatis et Cordis Extractum solution obtained from an autolysate of the liver and heart of the Barbary duck (Anas barbariae), diluted and potentised 200-fold using Korsakov’s method (200K). The same batch of sterile water (OTEC^®^ Laboratoires Aguettant, Lyon, France) was used as the vehicle in all the experiments. The homeopathic preparations were blinded and tested on cells to avoid experimental bias. The blinding was performed by a person not involved in the experiment. The decoding of the samples was revealed after the statistical analyses.

### 4.2. Primary Murine Microglial Cells

The protocol of microglial cell preparation and the cell seeding for the treatments were as previously described [25]. Briefly, murine microglial cells were collected from whole brains of postnatal day 0 mouse pups. After manual dissociation and several centrifugation steps at 200 g for 5 min, the brain cell pellet was resuspended in a cell culture medium and plated in T-75 flasks. Then, the cells were cultured for 20 days to obtain a culture enriched in microglial cells.

### 4.3. Atomic Force Microscopy (AFM) Sample Preparations

The cells were seeded on polyethyleneimine PEI-coated Nunc^®^ 35mm Petri dishes at a density of 250,000 cells/well. The next day, the entire culture medium was replaced to remove debris. Before adding the treatments, the stiffness of the cells was measured at T0. The cells were incubated with *A. barbariae* 200K (6% *v*/*v*), the vehicle (6% *v*/*v*) or JPK (25 nM), and the stiffness was measured after 1 min. The resting control cells corresponds to the condition without any treatment. All the experiments were performed in the complete culture medium at room temperature. The kinetic study of the cells aimed to investigate the cell stiffness, with AFM data recorded at intervals of 1 min, 2 min, 4 min, and 20 min after the addition of the treatment. Additionally, two other kinetics were conducted to identify the optimal time point for treatment-induced cell activation and the most responsive time point following treatment with *A. barbariae* 200K. For the LPS activation, five time points were examined at 2 h, 4 h, 6 h, 8 h, and 24 h compared to T0, with measurements taken randomly on ten cells. Similarly, the response time points for *A. barbariae* 200K treatment were assessed at 2 min, 4 min, and 20 min compared to T0, using the same number of cells.

AFM indentation experiments were carried out using a Resolve Bioscope (Bruker Nano Surface, Santa Barbara, CA, USA) that was mounted on an optical microscope (DMi8, Leica, Wetzlar, Germany) equipped with a 20× objective. A Nanoscope V controller with Nanoscope software version 8.15 was used. All the quantitative measurements were performed using standard pyramidal tips (ScanAsyst-Air, Bruker AFM probes, Inc., Camarillo, CA, USA). The tip radius given by the manufacturer was 8–12 nm. The spring constant of the cantilever was measured using the thermal tune method and was 0.4 N/m [44]. The deflection sensitivity of the cantilever was calibrated against a sapphire wafer. The Petri dish was positioned on an XY motorised stage and held using a magnetic clamp. Then, the AFM head was mounted on the stage. Only isolated cells were used for the AFM measurements. Each AFM experiment consisted of acquiring a matrix of force-indentation curves. For each condition, the study was conducted on 30 isolated cells. Force curves were acquired using different modes depending on the experiments.

For comprehensive cell measurements and time-lapse studies, force curves were obtained using the Ramp mode. Each AFM measurement entailed acquiring 100 force-indentation curves, sampled from a 10 × 10 µm grid with indentation points spaced 200 nm apart, all on a single cell. A force of 100 nN was applied at a rate of 5 Hz, with a sampling of 2048 points per curve.

### 4.4. Atomic Force Microscopy (AFM) Analysis

For the global measurement and kinetic study, we used the Hertz–Sneddon model to extract quantitative data on the elastic modulus from each curve, covering an indentation range of 0 to 1 µm. This model is particularly suitable when the indentation depth is smaller than the sample thickness, which applies here, as our cell samples typically ranged between 5 and 10 µm thick, and the indentation depth remained below 1 µm [45,46].

The Hertz–Sneddon model assumes a rigid cone indenting a flat surface, with the force (F) from the force curve given by the formula:(1)$$F=\frac{2}{\pi }\;.\;\;\frac{E}{1-\nu^{2}}\;.\;\tan\left(\alpha \right).\delta^{2}$$

F is the force from the force curve, E is the Young’s modulus, ν is the Poisson’s ratio, α is the half-angle of the indenter and δ is the indentation.

We assumed that our sample was perfectly incompressible, leading us to utilise a Poisson’s ratio of 0.5. However, due to the uncertainties regarding both the Poisson’s ratio and the exact shape of the tip, we only report an “apparent modulus” (Ea) in this study. Since the experimental conditions remained consistent across all the samples examined, Ea serves as a reliable metric for the comparisons with a high degree of confidence.

### 4.5. Confocal Microscopy

The cells were initially seeded onto PEI-coated Nunc^®^ 35 mm Petri dishes at a density of 250,000 cells per dish. Subsequently, treatments were administered following the protocol outlined for the AFM experiments. Post-treatment, the cells underwent a sequential procedure: first, they were rinsed once with phosphate-buffered saline (PBS) and then fixed for 10 min at room temperature with a 3.7% formaldehyde solution in PBS. This fixation was performed at different time points 1 or 4 min after the treatments, depending on the experiment.

After the fixation, the cells were washed twice for 5 min in PBS buffer. Subsequently, actin filaments were stained using 2 µM Phalloidin–Tetramethylrhodamine and the nucleus was stained with 2 µM 4′,6-diamidino-2-phenylindole (DAPI) from Sigma-Aldrich-Merck (Saint-Quentin-Fallavier, France). Following the staining, the cells were rinsed twice with PBS 1X solution, and the medium was replaced with a PBS/glycerol mixture (50/50). Finally, coverslips were placed onto the Petri dishes. A homemade 3D-printed support was used to hold the dish on the microscope, and images were acquired using an LSM 980 confocal microscope (Carl Zeiss, Rueil-Malmaison, France) equipped with a ×63 objective (Plan-Apochromat 63×/1.40 oil M27 with a numerical aperture of 1.40 and a working distance of 0.19 mm). Image quantification was performed using the open-source Image J software (version 1.54f).

### 4.6. Viability Assay

XTT reduction assays (Cell Proliferation kit II, Roche Diagnostics, Meylan, France) were carried out to investigate the cell viability, as previously described [25]. Microglial cells were seeded on a PEI-coated Nunc^®^ 96-well plate at the density of 50,000 cells/well. The next day, the cells were treated with 10 ng/mL LPS for 4 h and treated with *A. barbariae* 200K (1%, 3%, 6%, and 12% *v*/*v*) or the vehicle (1%, 3%, 6%, and 12% *v*/*v*) for 20 h. Four hours before the end of the treatment, XTT labelled mixture (0.3 mg/mL final concentration) was prepared in accordance with the manufacturer’s instructions and was added to each well. The absorbance at 465 nm (formazan) and at 650 nm (reference wavelength) was quantified using the SPARK 10M microplate reader (Tecan). The absorbance values obtained were used to determine the cell viability. Wells without cells were performed and used as blank references. LPS-treated cells were considered as viability references (viability ratio = 1).

### 4.7. Measurement of the Total and Mitochondria-Derived ROS Level

The level of reactive oxygen species (ROS) was assessed using two different cell-permeant dyes: CellROX™ Deep Red (Invitrogen, Illkirch, France) and MitoSOX™ Red (Invitrogen, Illkirch, France). The probes are non-fluorescent while in a reduced state and exhibit bright fluorescence when CellROX Deep red is oxidised by multiple reactive oxygen species and MitoSOX red with anion superoxide. Fluorescence provides a direct measure of the ROS levels. For the CellROX assay, the microglial cells were inflamed with LPS (10 ng/mL, *E. coli* strain O111:B4, Sigma-Aldrich-Merck, Saint-Quentin-Fallavier, France) for 4 h and treated with *A. barbariae* 200K (6% *v*/*v*), the vehicle (6% *v*/*v*) or dexamethasone (DEX 0.63 µM, Sigma-Aldrich-Merck, Saint-Quentin-Fallavier, France). After 6 h or 20 h of LPS, the cells were stained with 10 μM CellROX™ Deep Red reagent for 30 min. For the MitoSOX assay, the cells were incubated for 4 h with LPS and then co-incubated with the same treatments described above for 2 h, except the DEX concentration was at 2.9 µM instead of 0.63 µM. Before the end of the treatments, the MitoSOX™ Red probe (2.5 µM) was added for 10 min. At the end of the incubations, the cells were washed with Dulbecco’s Phosphate Buffered Saline (DPBS) and fixed with 3.7% formaldehyde solution (Sigma Aldrich-Merck F1635) in DPBS before image acquisition. For each well, at least 3 images were randomly acquired with identical exposure parameters using an inverted fluorescence microscope (Zeiss, Axio Observer Z1) equipped with a Hamamatsu camera (Orca Flash 4.0) and an LD Plan-Neofluar 20× objective. The quantification of fluorescence was performed using the open-source ImageJ software (version 1.50i). The intensity of the fluorescence was proportional to the total ROS or mitochondrial superoxide anion level. The results are expressed as a fractional change in the fluorescence intensity relative to the baseline in the untreated cells without LPS stimulation.

### 4.8. Statistical Analysis

#### 4.8.1. ROS

The statistical analyses were performed using GraphPad Prism 7 for Windows (GraphPad Software, San Diego, CA, USA). All the results are presented as the mean ± standard deviation (SD). Each result is representative of at least three independent experiments. The Shapiro–Wilk test was used to check if the analysed data were normally distributed. The Kruskal–Wallis test (nonparametric One-way ANOVA) was used to evaluate the statistical significance, as indicated by a * *p* < 0.05, ** *p* < 0.01, *** *p* < 0.001, **** *p* < 0.0001. A post hoc test was conducted using Tukey’s multiple comparisons test.

#### 4.8.2. AFM

The statistical analyses were conducted on all the experimental data using R Studio. The normality was assessed using the Shapiro–Wilk test, while the homogeneity of variances was assessed using the Levene test. The comparisons of means were made using either the *t*-test or the Wilcoxon test, with significance levels denoted as follows: * *p* < 0.05; ** *p* < 0.005; *** *p*< 0.0005.

## Figures and Tables

**Figure 1 ijms-26-01451-f001:**
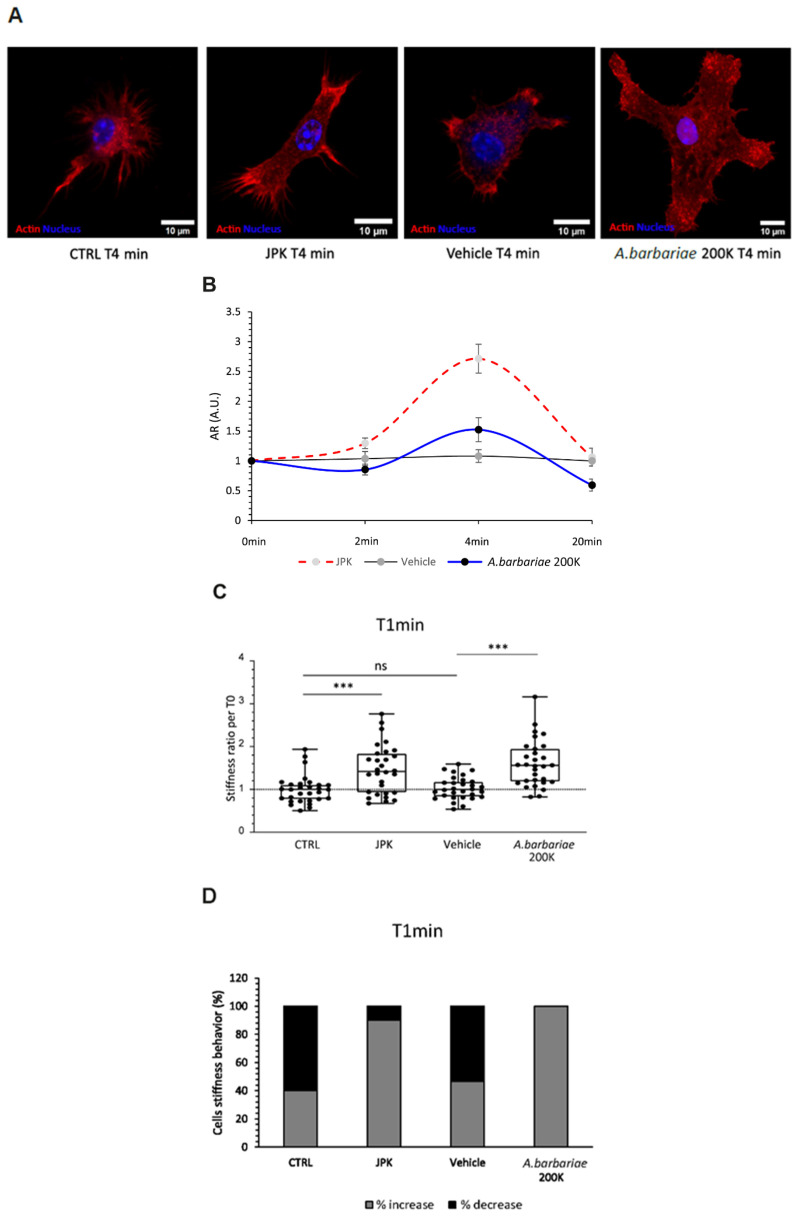
Effect of treatment on morphological and mechanical properties in resting primary murine microglial cells. (**A**) Representative confocal images of resting microglial cell after different treatments: control cell (CTRL), *A. barbariae* 200K, vehicle, and JPK at 4 min. (**B**) The Box and Whiskers plot representation illustrates the aspect ratio over time under various treatments. (**C**) Stiffness measurements were taken in cells at baseline (T0) and 1 min after the addition of a solution (JPK, vehicle, *A. barbariae* 200K) or no addition (CTRL). The graph displays the ratio of values after and before the solution was added for each condition. A ratio greater than 1 indicates an increase in stiffness (Ea) following the addition, while a ratio less than 1 suggests a decrease in stiffness. The lines across the boxes represent median values, and the boxes themselves indicate the interquartile ranges (IQRs). The dots represent individual data points. Statistical significance: *** *p* < 0.0005, ns = not significant. (**D**) Proportion of cells reacting in increasing (light grey) or decreasing (black) their elastic modulus after addition. (**E**) Cell stiffness monitored over time before (T0) and after addition or not of treatment. All data were normalised to the T0 value. These blind experiments were repeated three times in an independent manner. (CTRL: resting control cells; JPK: jasplakinolide; vehicle: sterile water, *A. barbariae*: *Anas barbariae*).

**Figure 2 ijms-26-01451-f002:**
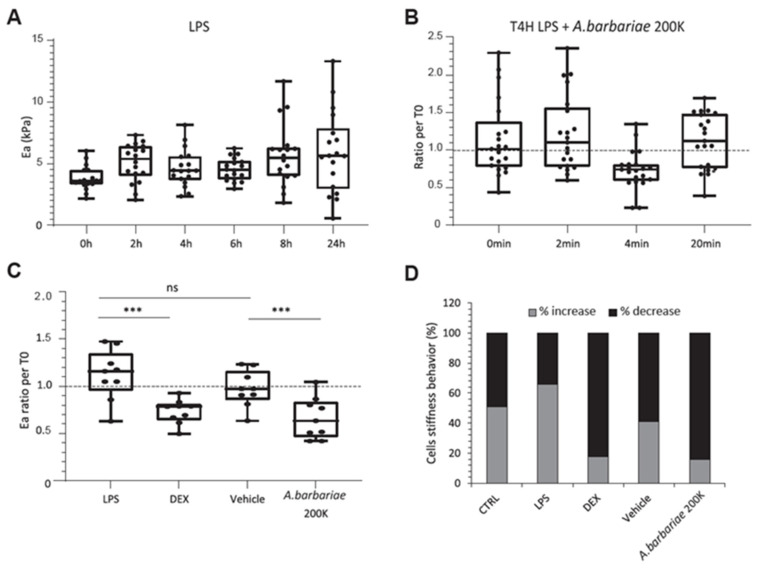
Effect of LPS and *A. barbariae* 200K on mechanical properties in inflamed primary murine microglial cells. (**A**) Observation of the mechanical response of cells over time following LPS stimulation only. (**B**) Graph showing the biomechanical changes of LPS inflamed cells (4 h) treated with *A. barbariae* 200K at different times. (**C**) Mechanical responses of LPS-treated cells (4 h) depending on various treatments after 4 min. The Box and Whiskers plot was chosen to represent these data’s distribution. The dots indicate individual data points. The lines across the boxes represent the median values, while the box itself spans the interquartile range (IQR). The dotted line at 1 represents the untreated cells. Statistical significance: *** *p* < 0.0005, ns = not significant. (**D**) Histogram of the number of cells responding positively (light grey) or negatively (black) to the treatment. These blind experiments were repeated three times in an independent manner. (CTRL: resting control cells; LPS: lipopolysaccharide; DEX: dexamethasone; vehicle: sterile water; *A. barbariae*: *Anas barbariae*).

**Figure 3 ijms-26-01451-f003:**
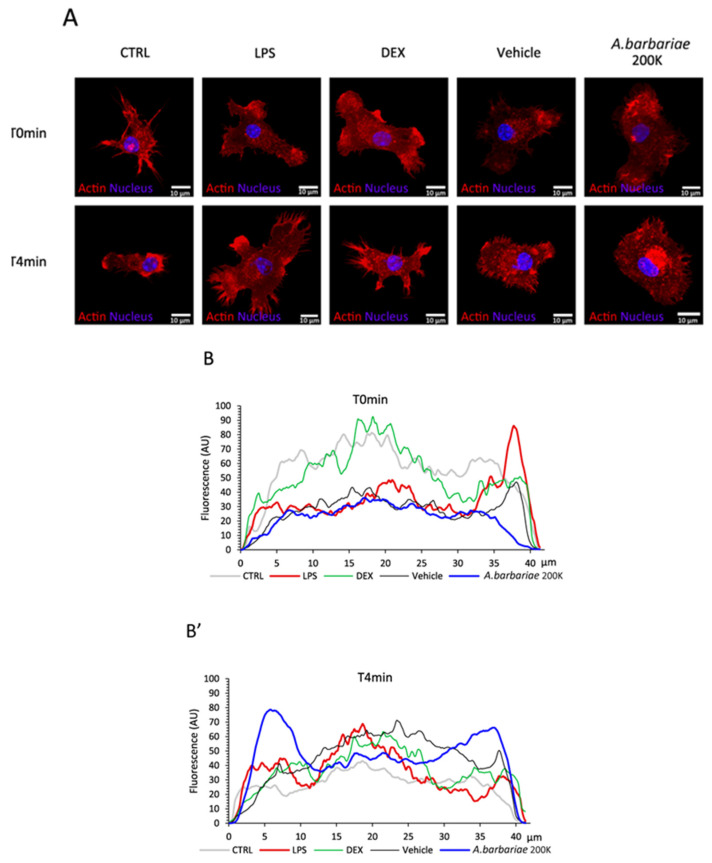
Effect of *A. barbariae* 200K on morphology in inflamed primary murine microglial cells. (**A**) Confocal images of filamentous actin in red and the nucleus in blue allow for a qualitative analysis of cell morphology. (**B**,**B’**) Analysis of filamentous actin distribution within the cell. The intensity profile was generated along the longest axis of the cell passing through the nucleus centre. Cell sizes are normalised for graphical overlay. The profile represents an average profile of 10 cells. These blind experiments were repeated three times in an independent manner. (CTRL: resting control cells; LPS: lipopolysaccharide; DEX: dexamethasone; vehicle: sterile water; *A. barbariae*: *Anas barbariae*).

**Figure 4 ijms-26-01451-f004:**
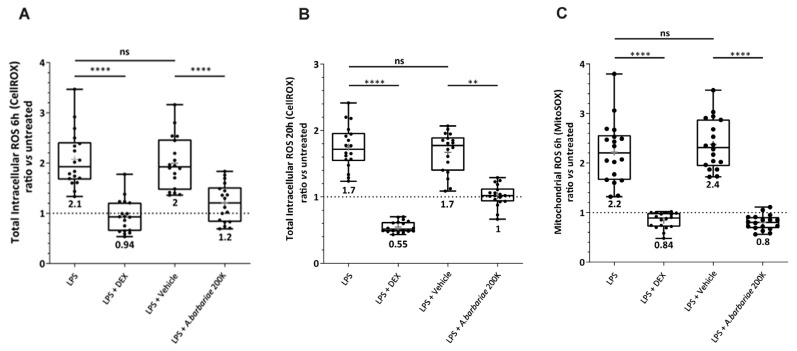
Effect of *A. barbariae* 200K on oxidative stress in inflamed primary murine microglial cells. The production of ROS was assessed using both CellROX and MitoSOX probes. These probes measured multiple reactive oxygen species produced in the cytoplasm (total intracellular content) with CellROX Deep Red probe and anion superoxide in the mitochondria with MitoSOX Red probe. The Box and Whiskers plot representation shows the antioxidant properties of *A. barbariae* 200K. (**A**,**B**) Total intracellular ROS measured using the CellROX probe (6 h and 20 h). (**C**) Mitochondrial ROS measured using the MitoSOX probe (6 h). All the values were divided by the fluorescence value of the untreated cells. The dotted line at 1 corresponds to the untreated cells. The Box and Whiskers plot was chosen to represent these data’s distribution. Lines across the boxes show median values, while the box itself spans the interquartile range (IQR). The dots represent individual value. The cross (grey) is the mean value displayed below the Box and Whiskers plot. Statistical significance: ** *p* < 0.01, **** *p* < 0.0001; ns = not significant. The blind experiments were repeated three times in an independent manner. (ROS: reactive oxygen species; LPS: lipopolysaccharide; DEX: dexamethasone; vehicle: sterile water; *A. barbariae*: *Anas barbariae*).

## Data Availability

Data will be made available on request.

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
