# Peer review of "Anas barbariae 200K Modulates Cell Stiffness and Oxidative Stress in Microglial Cells In Vitro"

_ijms, 2025, doi:10.3390/ijms26041451_

Round 1
Reviewer 1 Report
Comments and Suggestions for Authors
This paper hass numerous scientific shortcomings and lack of rigor. The main issue lies in the attempt to attribute significant cellular effects to a homeopathic preparation, which inherently poses a major limitation given the extremely diluted nature of such remedies. The claimed effects of Anas barbariae 200K on microglial cells lack plausibility from a pharmacological standpoint, as the dilution level makes it highly unlikely that any active ingredient remains in the solution. Thus, the observed changes in cell stiffness and ROS production are more likely the result of experimental bias rather than true biological responses. The authors also fail to provide a convincing mechanism of action, further diminishing the credibility of their conclusions. In summary, the use of a homeopathic remedy to induce measurable effects in glial cells is not scientifically substantiated, and this undermines the validity of the entire study.
For all the above reasons, I regret to believe that this paper does not meet the standards for publication in IJMS.
Comments on the Quality of English LanguageThe english is good, minor editing is required
Author Response
Comment 1: "This paper hass numerous scientific shortcomings and lack of rigor. The main issue lies in the attempt to attribute significant cellular effects to a homeopathic preparation, which inherently poses a major limitation given the extremely diluted nature of such remedies. The claimed effects of Anas barbariae 200K on microglial cells lack plausibility from a pharmacological standpoint, as the dilution level makes it highly unlikely that any active ingredient remains in the solution."
Response 1: Thank you for your time and your comments. We have amended the manuscript .
First of all, in the manuscript, all the changes appear highlighted in red for an addition and crossed out in grey for a removal. To be more convincing in this difficult scientific field, we improved the manuscript according to your comments.
- Experiments were blinded and repeated three times in an independent manner. The person who incubated the homeopathic preparation did not know the samples during all experiments. This has been added and made clear in the legends of the figures (line 122; 210-211; 245, 292-293) in addition to the Materials & Methods section (line: 360-362) . Furthermore, the decoding of the samples was revealed after the statistical analyses (line: 392-394).
- All these data were carefully statistically analyzed (normality, homoscedasticity, one-way ANOVA parametric or non-parametric, suitable post-hoc tests) and mentioned in Materials and Methods to check the statistically biological effects of the different compounds and controls (line: 516-527).
Comment 2: "Thus, the observed changes in cell stiffness and ROS production are more likely the result of experimental bias rather than true biological responses. The authors also fail to provide a convincing mechanism of action, further diminishing the credibility of their conclusions. In summary, the use of a homeopathic remedy to induce measurable effects in glial cells is not scientifically substantiated, and this undermines the validity of the entire study."
Response 2:
- Experimental bias from the experiments are excluded since the cell model has been used and several times published by others as mentioned in the discussion [1–3]. Another list of bibliography reference is added at the end.
- The sterile water used as the vehicle in all the experiments has the same batch number of water used to obtain A. barbariae 200K
- The same homeopathic dilutions and vehicle were utilized for the treatments for ROS production (CellROX, MitoSOX), cell stiffness and actin reorganization.
- To respond to reviewer 2, a dose dependence experiments with CellROX (repeated 3 times in an independent manner) were performed as complementary experiments fully demonstrating the credibility of our results
- The homeopathic dilutions were rigorously prepared by Laboratoires Boiron in compliance with the European Pharmacopoeia (Ph. Eur.) monographs according to the monograph 2371 and 1038 [4,5].
- We significantly observed a decrease of ROS production and cell stiffness in inflamed microglial cells with barbariae 200K. On the contrary, this effect was not observed with the vehicle. These results obtained with A. barbariae 200K are specific and statistically significant compared with vehicle.
- To emphasize the results in the discussion, we added the fact that we observed the same results on ROS production and cell stiffness with a different batches of barbariae 200K and vehicle. In our previous study, regarding ROS production, the treatments were different. The cells were incubated with A. barbariae 200K before the inflammation with LPS whereas here A. barbariae 200K was incubated after LPS-inflammation. In the present study, cell stiffness was assessed on both resting and LPS-inflamed cells with A. barbariae 200K (line: 288-294) . The results showed that A. barbariae 200K increased cell stiffness on resting cells and decreased it on inflamed cells.
Comment 3: "The authors also fail to provide a convincing mechanism of action, further diminishing the credibility of their conclusions."
Response 3: Unfortunately, the mechanism of action is not well identified for homeopathic preparations. In the literature, you find scientific evidence that homeopathic preparations have an effect on cells: ROS production, cell stiffness, membrane organization, neurite outgrowth, bioenergetic, gene expression, DNA damage and inflammation [6–12].
Additional clarifications: In the discussion, we added some explanations about our previous study and the present study (line: 320-327)
References
- dos Santos Pereira, M.; Maitan Santos, B.; Gimenez, R.; Guimarães, F.S.; Raisman-Vozari, R.; Del Bel, E.; Michel, P.P. The Two Synthetic Cannabinoid Compounds 4′-F-CBD and HU-910 Efficiently Restrain Inflammatory Responses of Brain Microglia and Astrocytes. Glia 2024, 72, 529–545, doi:10.1002/glia.24489.
- dos‐Santos‐Pereira, M.; Guimarães, F.S.; Del‐Bel, E.; Raisman‐Vozari, R.; Michel, P.P. Cannabidiol Prevents LPS‐induced Microglial Inflammation by Inhibiting ROS/NF‐κB‐dependent Signaling and Glucose Consumption. Glia 2020, 68, 561–573, doi:10.1002/glia.23738.
- Sepulveda-Diaz, J.E.; Ouidja, M.O.; Socias, S.B.; Hamadat, S.; Guerreiro, S.; Raisman-Vozari, R.; Michel, P.P. A Simplified Approach for Efficient Isolation of Functional Microglial Cells: Application for Modeling Neuroinflammatory Responses in Vitro. Glia 2016, 64, 1912–1924, doi:10.1002/glia.23032.
- Monograph 1038 (Homeopathic Preparations). In European Pharmacopoeia; EDQM Council of Europe Editions: Strasbourg, France, 2024.
- Monograph 2371 (Methods of Preparation of Homoeopathic Stocks and Potentisation). In European Pharmacopoeia; EDQM Council of Europe Editions: Strasbourg, France, 2024.
- Paumier, A.; Verre, J.; Tribolo, S.; Boujedaini, N. Anti-Oxidant Effect of High Dilutions of Arnica Montana, Arsenicum Album, and Lachesis Mutus in Microglial Cells in Vitro. Dose Response 2022, 20, 1–7, doi:10.1177/15593258221103698.
- Verre, J.; Boisson, M.; Paumier, A.; Tribolo, S.; Boujedaini, N. Anti-Inflammatory Effects of Arnica Montana (Mother Tincture and Homeopathic Dilutions) in Various Cell Models. J Ethnopharmacol 2024, 318, 117064, doi:10.1016/j.jep.2023.117064.
- Olioso, D.; Marzotto, M.; Bonafini, C.; Brizzi, M.; Bellavite, P. Arnica Montana Effects on Gene Expression in a Human Macrophage Cell Line. Evaluation by Quantitative Real-Time PCR. Homeopathy 2016, 105, 131–147, doi:10.1016/j.homp.2016.02.001.
- Fuselier, C.; Dufay, E.; Berquand, A.; Terryn, C.; Bonnomet, A.; Molinari, M.; Martiny, L.; Schneider, C. Dynamized Ultra-Low Dilution of Ruta Graveolens Disrupts Plasma Membrane Organization and Decreases Migration of Melanoma Cancer Cell. Cell Adh Migr 2023, 17, 1–13, doi:10.1080/19336918.2022.2154732.
- Lejri, I.; Grimm, A.; Trempat, P.; Boujedaini, N.; Eckert, A. Gelsemium Low Doses Increases Bioenergetics and Neurite Outgrowth. AJBIO 2022, 10, 51, doi:10.11648/j.ajbio.20221002.13.
- Lejri, I.; Grimm, A.; Trempat, P.; Boujedaini, N.; Eckert, A. Gelsemium Low Doses Protect against Serum Deprivation-Induced Stress on Mitochondria in Neuronal Cells. Journal of Ethnopharmacology 2025, 336, 118714, doi:10.1016/j.jep.2024.118714.
- Toma, C.-C.; Marrelli, M.; Puticiu, M.; Conforti, F.; Statti, G. Effects of Arnica Phytotherapeutic and Homeopathic Formulations on Traumatic Injuries and Inflammatory Conditions: A Systematic Review. Plants 2024, 13, 3112, doi:10.3390/plants13213112.
Reviewer 2 Report
Comments and Suggestions for Authors
The manuscript is well written and the objectives of the work are described.
However, the manuscript needs improvement in the methods part. In particular, attention should be paid to acronyms and indication of instrumentation. For example section 4.3. AFM sample preparation.
The graphs need to be improved to make them more understandable and readable.
Doses and dilutions used in cell treatment are not clear, and graphs of dose dependence and cell viability are not shown.
The manuscript needs to be improved and revised after careful review to reach the standards required by the journal
Author Response
Comment 1: "However, the manuscript needs improvement in the methods part. In particular, attention should be paid to acronyms and indication of instrumentation. For example section 4.3. AFM sample preparation."
Response 1:
- We acknowledged your comments, and we have amended the manuscript accordingly to improve it. In the manuscript, all the changes appear highlighted in red for an addition and crossed out in grey for a removal.
- The Materials and Methods part was improved for a better understanding. The final concentrations of DAPI (line: 469) and phalloidin (line: 468) were now specified in the Materials & Methods. The abbreviations of PEI=polyethyleneimine (line: 408), DAPI=4′,6-diamidino-2-phenylindole (line: 469), Dulbecco’s Phosphate Buffered Saline=DPBS (line: 505) were added in the manuscript.
- For more clarity, we used ‘vehicle’ for sterile water (line: 74, 123, 270) to avoid the confusion with CTRL which corresponds to the ‘resting control cells’ (line: 73, 122, 132, 211, 245, 412) where the same volume of cell culture medium was added to the microglial cells to reproduce the addition of homeopathic medicine A. barbariae 200K.
Comment 2: "The graphs need to be improved to make them more understandable and readable."
Response 2:
- The legends of the figures have been improved by adding the abbreviations of the treatments and to notify that the experiments were blinded performed and repeated three times, the sentence “The blind experiments were repeated three times in an independent manner” is now added (line: 122-123, 210-211, 245-2246, 292-293, 392-393).
- The figures were improved. In the journal format, figure 1 was too small. The size has been increased. To better distinguish the treatments in fig1B and E and fig3B and B’, the curves now appeared in color .
- Regarding Fig4 with ROS production, we are sorry but the legends on the y-axis were not correct. I agree. The legend on y-axis should be:
- Fig3A Total Intracellular ROS 6 h (CellROX)
- Fig3B Total Intracellular ROS 20 h (CellROX)
- Fig3C Mitochondrial ROS 6h (MitoSOX)
They have been modified in the new figure 3.
Comment 3: "Doses and dilutions used in cell treatment are not clear, and graphs of dose dependence and cell viability are not shown."
Response 3:
- The final concentrations of DAPI (line: 469) and phalloidin (line: 468) were now specified in the Materials & Methods. We also specified the concentrations of LPS (line: 263) and dexamethasone (line: 267)
- To reply to your comment about the cell viability, the viability of the cells was monitored using XTT viability assay (repeated 3 times in an independent manner). The microglial cells were incubated with LPS (10 ng/mL) and barbariae 200K or vehicle with different doses at 1%, 3%, 6% and 12% (v/v). The results show that all the treatments do not affect the cell viability. The cell viability results reinforce the fact that all the incubations were performed with 6% (v/v) of A. barbariae 200K. The figure S1 is added to supplemental data. The viability assay and the results have been added in the Materials and Methods (line: 478-489) and Results (line: 249-252) sections, respectively. The paragraph number in Materials and Methods has been added in ‘4.6 Viability assay’ (line: 478)
- To reply to dose dependence comment, we performed complementary experiments with CellROX (repeated 3 times in an independent manner) by testing different doses at 1-3-6-12% (v/v) of barbariae 200K or vehicle. With vehicle 1%, 3% and 6%, all the values are similar compared to LPS. In contrast, the vehicle 12% (v/v) is affecting ROS production. This is the reason why in all experiments we used 6% of homeopathic preparation with cells. The figure S2 is added to supplemental data. Those results are mentioned in the manuscript (results section, line: 253-256).
- The conclusion has been moved and added just after the discussion section (line: 378-382).
Additional clarifications
In the discussion, we added some explanations about our previous study and the present study (line: 320-327)
Altogether, those new data and experiments reinforce our results demonstrating a biological effect of A. barbariae 200K in this microglial cell model.
Round 2
Reviewer 1 Report
Comments and Suggestions for Authors
The authors well replied to my previous comments.
Now, I suggest only to insert the limitations of the study (based on my previous comments and you response) at the end of the Discussion.
Author Response
We acknowledged your comment, and we have amended the manuscript.
All the changes appear highlighted in green (round 2) or in red from round 1 for an addition and crossed out in grey for a removal.
Comment 1: "Now, I suggest only to insert the limitations of the study (based on my previous comments and you response) at the end of the Discussion"
Response 1:
We agree with your suggestion to add some of our answers from your first comments (round 1). In the discussion, we have added a paragraph about the mechanism of action of homeopathic preparations which is not yet identified but homeopathic preparations have a biological effect (line 384-388). The bibliography has been updated by adding references n° 38 (Olioso et al, 2016), 39 (Lejri et al, 2022), 40 (Lejri et al, 2025), and 41 (Toma et al., 2024).
In Materials and Methods, we have added the references of the monographs for homeopathic preparations (line: 397-398) ref n°: 42 and 43
We thought that these two points were important to mention in the manuscript.
All the other comments from response 1 were already in the manuscript. For example, the blinding of the sample was already notified in the Materials and Methods and in the legends of the figures. Therefore, we did not think necessary to write this again in the discussion.
In the Discussion, we added sentences about the redistribution of actin in inflamed microglial cells (line: 371-375).
All your comments in round 1 and 2 were taken into account and helped to improve the manuscript.
Reviewer 2 Report
Comments and Suggestions for Authors
The authors responded to most of the comments and revised the manuscript accordingly.
The authors responded to most of the comments and revised the manuscript accordingly.
It is evident that the manuscript contains notes on AI, and that certain charts bear the annotation 'description générée automatiquement'. Consequently, it is requested that the manuscript be subject to a quality control check prior to its publication.
Author Response
We acknowledged your comments. We have amended the manuscript accordingly to improve it. All the changes appear highlighted in green (round 2) or in red (round 1) for an addition, and crossed out in grey for a removal.
Comments 1:" It is evident that the manuscript contains notes on AI, and that certain charts bear the annotation 'description générée automatiquement'. Consequently, it is requested that the manuscript be subject to a quality control check prior to its publication."
Response 1:
We are very surprised about your comment using AI. We did not use AI to write the paper, to make the figures or the graphical abstract.
So, we investigated where this notification came from. We sincerely think it is a misunderstanding and mistranslation of French since your comment 'description générée automatiquement' is written in French.
As numerous labs over the world, we are using Microsoft 365. In Word, there is a function to tick for generating automatic description by default. This option suggests text automatically while typing but it is absolutely not AI.
You will find this function (see screenshot below). In English, it is written "Automatic ALT Text". This function was ticked.
To be fully transparent, the graphs were generated in Excel or GraphPad Prism from the raw data or normalized and they were imported in Adobe Illustrator. The images were analysed with Fiji software.
At last and in order to convince you, we analyzed the manuscript through Copilot for checking if AI was utilized and the answer is 'No'.
We also noticed from your first comments that you did not improve the marking after we made the changes in the manuscript even if you wrote "The authors responded to most of the comments and revised the manuscript accordingly'.
Nonetheless, we took into account your marking and improved the manuscript on the following points:
- Regarding the graphical abstract, we added in the Acknowledgements that Servier Medical Art was partialy used to create the cell images (line : 563-564).
- In Materials and Methods, we have checked the acronym such as AFM: Atomic Force Microscopy (line: 418, 452). The other acronyms are explained in previous parts of the paper as mandatory by the Author's guidelines of IJMS.
- In the Discussion, we added sentences about the redistribution of actin in inflamed microglial cells (line: 371-375).
We sincerely hope that all the changes in the manuscript correspond to your expectation.
Round 3
Reviewer 1 Report
Comments and Suggestions for Authors
Nothing to add
Comments on the Quality of English LanguageMinor editing is required